# History of the Study of the Genus *Thiothrix*: From the First Enrichment Cultures to Pangenomic Analysis

**DOI:** 10.3390/ijms23179531

**Published:** 2022-08-23

**Authors:** Nikolai V. Ravin, Tatyana S. Rudenko, Dmitry D. Smolyakov, Alexey V. Beletsky, Maria V. Gureeva, Olga S. Samylina, Margarita Yu. Grabovich

**Affiliations:** 1Institute of Bioengineering, Research Center of Biotechnology, Russian Academy of Sciences, Prospect 60-letiya Oktyabrya 7/1, 119071 Moscow, Russia; 2Department of Biochemistry and Cell Physiology, Voronezh State University, Universitetskaya pl., 1, 394018 Voronezh, Russia; 3Winogradsky Institute of Microbiology, Research Centre for Biotechnology, Russian Academy of Sciences, Prospect 60-letiya Oktyabrya 7/2, 117312 Moscow, Russia

**Keywords:** *Thiothrix*, phylogeny, pangenome, metagenome-assembled genome

## Abstract

Representatives of the genus *Thiothrix* are filamentous, sulfur-oxidizing bacteria found in flowing waters with counter-oriented sulfide and oxygen gradients. They were first described at the end of the 19th century, but the first pure cultures of this species only became available 100 years later. An increase in the number of described *Thiothrix* species at the beginning of the 21st century shows that the classical phylogenetic marker, 16S rRNA gene, is not informative for species differentiation, which is possible based on genome analysis. Pangenome analysis of the genus *Thiothrix* showed that the core genome includes genes for dissimilatory sulfur metabolism and central metabolic pathways, namely the Krebs cycle, Embden–Meyerhof–Parnas pathway, glyoxylate cycle, Calvin–Benson–Bassham cycle, and genes for phosphorus metabolism and amination. The shell part of the pangenome includes genes for dissimilatory nitrogen metabolism and nitrogen fixation, for respiration with thiosulfate. The dispensable genome comprises genes predicted to encode mainly hypothetical proteins, transporters, transcription regulators, methyltransferases, transposases, and toxin–antitoxin systems.

## 1. Introduction

The first attempts to describe bacteria from the genus *Thiothrix* were made by Rabenhorst as early as 1865, when he described the first member of this genus as *Beggiatoa nivea* [1]. Winogradsky, in turn, based on studies of key features of enrichment culture, named a new genus, *Thiothrix* [2]. The genus *Thiothrix* belongs to the class *Gammaproteobacteria*, order *Thiotrichales*, family *Thiotrichaceae*.

Habitats of *Thiothrix* vary from natural sulfidic waters, irrigation systems, and activated sludge in wastewater treatment plants to ectosymbionts of invertebrates in deep-sea hydrotherms. The influx of H_2_S into the growth zone of these bacteria occurs from sulfidic springs, either from the near-bottom layers of sediments (in the shallow waters of lakes, in ponds, sea littorals, etc.) or from hydrothermal vents [3,4]. The hydrogen sulfide concentration can vary significantly—from tens of micrograms to several milligrams per litre. In nature, *Thiothrix* forms powerful foulings, visible to the naked eye.

Currently, the genus *Thiothrix* includes aerobic and facultative anaerobic, attached, filamentous, non-motile bacteria. They are capable of auto- and heterotrophic growth and are characterized by a respiratory type of metabolism. During autotrophic growth, CO_2_ fixation occurs through the Calvin–Benson–Bassham cycle. Their ribulose-1,5-bisphosphate carboxylase-oxygenase (RuBisCO) belongs to types IAq, IAc, and II. All genomes contain genes encoding all enzymes of the Krebs cycle, with the exception of malate dehydrogenase (MDH), which is functionally replaced by malate:quinone oxidoreductase (MQO). *Thiothrix* spp. are capable of organotrophic growth, as well as lithotrophic growth in the presence of reduced sulfur compounds. During lithotrophic growth in the presence of hydrogen sulfide and thiosulfate, elemental sulfur is accumulated intracellularly. Hydrogen sulfide is oxidized to sulfur by the sulfide:quinone oxidoreductase (SQR) and flavocytochrome *c*-sulfide dehydrogenase (FCSD). Thiosulfate is oxidized by the branched sulfur-oxidizing system (SOX) pathway without SoxCD with the formation of sulfur and sulfate. Sulfite is oxidized via direct (membrane-bound cytoplasmic sulfite:quinone oxidoreductase (SoeABC) and indirect (adenosine phosphosulfate reductase (AprAB) and ATP sulfurylase (Sat)) oxidation pathways (Figure 1).

Before 1965, eleven morphotypes of the genus *Thiothrix* were described mainly in natural marine and freshwater habitats containing hydrogen sulfide [5,6,7]. These microorganisms are differentiated based solely on the diameter of the filaments and the characteristics of the habitat. Subsequently, obtaining pure cultures made it possible to reveal that the morphology of the genus *Thiothrix* is variable [8]. The final invalidity of using phenotypic characters for the taxonomy of the genus *Thiothrix* was confirmed by Howarth et al., 1999 [9]. In 1983, Larkin and Shinabarger isolated the first pure culture for a representative of this genus [5]. Based on Winogradsky’s description, Shinabarger suggested that the culture he received was *Thiothrix nivea*. The strain JP2 with a validly published name *Thiothrix nivea* JP2^T^ (=ATCC 35100^T^ = DSM 5205^T^) is the only established neotype of the species [5].

At the end of the 20th century, several new isolates were obtained: *Thiothrix ramosa* [10], *Thiothrix arctophila* [11], and *Thiothrix* sp. CT3 [12]. Unfortunately, two proposed species, *T*. *ramosa* [10] and *T*. *arctophila* [11], are absent in international collections and were lost (Dubinina, personal communication).

The genus *Thiothrix* was significantly expanded by Howarth in 1999 [9]. Four new species were included in the genus: *Thiothrix fructosivorans*, *Thiothrix unzii*, *Thiothrix defluvii*, and *Thiothrix eikelboomii*. The last two species were assigned to the Eikelboom type 021N group within the genus *Thiothrix*, and the species *T. nivea*, *T. fructosivorans*, and *T. unzii* were assigned to the *T. nivea* group [9]. Comparative analysis of the 16S rRNA gene sequence of members of the Eikelboom type 021N and the *T. nivea* groups showed low similarity (90–91%). However, the notable phenotypic similarity between the Eikelboom type 021N group and the *Thiothrix nivea* group did not allow division into new genera at that time.

Two extra representatives of Eikelboom type 021N, *Thiothrix disciformis* and *Thiothrix flexilis*, and two species from the *Thiothrix nivea* group, *Thiothrix lacustris* and *Thiothrix caldifontis*, were described in later years [13,14].

The increase in the number of species of the genus *Thiothrix* has set the task of searching for new phylogenetic markers. Pure cultures of the genus *Thiothrix* isolated from various biotopes (hydrogen sulfide springs, wastewater treatment plants, the White Sea littoral, activated sludge treatment systems, freshwater lakes, groundwater, invertebrate ectosymbionts, etc.) have a similar morphotype, but a rather variable metabolism. The phylogeny based on the 16S rRNA gene sequences does not always correspond to the phylogenetic diversity of the representatives of this group [8]. Some strains assigned to the same species based on the 16S rRNA gene were reclassified as separate species after determination of whole-genome sequences (*T. lacustris* BL^T^, ‘*Thiothrix litoralis*’ AS^T^, and ‘*Thiothrix winogradskyi*’ CT3^T^) (Figure 2).

However, the 16S rRNA gene can be successfully used to identify *Thiothrix* at the genus level since the levels of 16S rRNA gene sequence identity between representatives of the genus *Thiothrix* exceed 94%, while with members of other genera, this value is below 91%.

## 2. Genome-Based Phylogeny

The determination of complete genome sequences for *T. disciformis*, *T. eikelboomii*, *T. flexilis*, *T. caldifontis*, *T. lacustris*, and *T. nivea* has enabled a more accurate phylogenetic analysis. In 2018, Boden and Scott undertook a multi-phase study which included morphological, biochemical, physiological, and genomic properties, and gene-based phylogeny to reclassify *Thiothrix* species. The 16S rRNA gene (*rrs*), recombination protein A (*recA*), polynucleotide nucleotide transferase (*pnp*), translation initiation factor IF-2 (*infB*), glyceraldehyde-3-phosphate dehydrogenase (*gapA*), glutamyl-tRNA synthetase (*glnS*), elongation factor EF-G (*fusA*), and concatenated sequences of 53 ribosomal proteins allowed the distribution of *Thiothrix* species between three different families: *Thiolineaceae*, *Thiofilaceae*, and *Thiotrichaceae* [15].

*Thiothrix defluvii* and *Thiothrix flexilis* were reclassified as representatives of the new genus *Thiofilum* within the family *Thiofilaceae* with the proposed names *Thiofilum flexile* and *Thiofilum defluvii*. *Thiothrix eikelboomii* and *Thiothrix disciformis* were placed in the new genus *Thiolinea* within the new family *Thiolineaceae* with the proposed names ‘*Thiolinea eikelboomii*’ for *Thiothrix eikelboomii*. However, the reclassification of *Thiolinea eikelboomi*i is currently only formal due to the lack of cultures in two international collections, as required for species validation. *T. caldifontis*, *T. lacustris*, *T. nivea*, *T. unzii*, and *T. fructosivorans *remained in the genus *Thiothrix* [15].

The development of genomics and metagenomics methods made it possible to obtain complete genome sequences and use them for phylogenetic studies, which, in turn, contributed to the development of a new genome-based taxonomic system of prokaryotes [16]. The whole-genome comparison has higher accuracy and resolution than taxonomy based on individual phylogenetic markers. Whole-genome sequences of isolates *T. fructosivorans* Q^T^, *T. unzii* A1^T^, *Thiothrix litoralis* AS^T^, ‘*Thiothrix subterranea*’ Ku-5^T^ [8], and ‘*Thiothrix winogradskyi*’ CT3^T^ [17], as well as metagenome-assembled genomes (MAGs) of ‘*Candidatus* Thiothrix anitrata’ A52, *Candidatus* Thiothrix moscovensis RT [18,19], and *Candidatus* Thiothrix singaporensis SSD2 [19,20], were obtained during the last three years. Just recently, ‘*Candidatus* Thiothrix sulfatifontis’ KT was obtained from the fouling of a hydrogen sulfide source [17].

The main characteristics of the obtained genomes are shown in Table 1.

Several MAGs were obtained from the Svalliden-Norrby groundwater metagenome, Oskarshamn, Sweden. When analyzing the obtained MAGs, we found that the genome assemblies GCA_018813855.1, GCA_018822845.1, and GCA_018825285.1 represented members of the genus *Thiothrix*. ANI values between the three genomes were 99.8–100%, which indicates that all assemblies represented the same species. For analysis of the pangenome assembly, GCA_018813855.1 (Modern_marine.mb.207), designated as MAG of *Thiothrix* sp. 207, was chosen. ANI and dDDH values between MAG of *Thiothrix* sp. 207 and other representatives of the genus (76–79% and 20–26%, respectively) indicated that this genome represented a novel *Candidatus* species (Figure 2).

## 3. Pangenome Analysis

Based on the genome sequences of eight isolates and five MAGs, an updated phylogeny of the genus *Thiothrix* was proposed (Figure 3).

The availability of a significant number of whole-genome sequences enabled a pangenome analysis of the genus *Thiothrix*. All genome sequences were of high quality except for the MAG of ‘*Ca.* Thiothrix singaporensis’ SSD2, which contained multiple frameshifts [8] and therefore was excluded from comparison. The completeness of four other MAGs according to CheckM estimates was above 98%. Pangenome analysis of 12 species, ‘*Thiothrix*
*winogradskyi*’ CT3^T^ (GCF_021650945.1), *T. lacustris* BL^T^ (GCF_000621325.1), *T.*
*litoralis* AS^T^ (GCF_017901135.1), ‘*Thiothrix subterranea’* Ku-5^T^ (GCF_016772315.1), *T. caldifontis* G1^T^ (GCF_900107695.1), *T. unzii* A1^T^ (GCA_017901175.1), *T. nivea* JP2^T^ (GCF_000260135.1), *T. fructosivorans* Q^T^ (GCA_017349355.1), *Ca.* Thiothrix moscovensis RT (GCA_016292235.1), ‘*Ca*. Thiothrix anitrata’ A52 (GCF_017901155.1), ‘*Ca.* Thiothrix sulfatifontis’ KT (GCA_022828425.1), and MAG of *Thiothrix* sp. 207 (GCA_018813855.1), showed that the pangenome of the genus *Thiothrix* includes 13,436 gene clusters. The core genome contains 1271 genes, and the auxiliary genome includes 4811 gene clusters present in 2 to 11 genomes. The number of unique genes varies in different species from 359 to 1215 (Figure 4).

The genus *Thiothrix* accommodates lithotrophic sulfur-oxidizing bacteria possessing various systems for the oxidation of reduced sulfur compounds. Bacteria from the genus *Thiothrix* are able to use a number of reduced inorganic sulfur compounds as an energy source, for example, hydrogen sulfide, sulfur, sulfite, and thiosulfate. Pangenome analysis showed the presence of two types of SQR (SqrF and SqrA) in the core genome. The FCSD complex is also present in all *Thiothrix* genomes but due to sequence divergence appeared to be among the auxiliary genes. Genes of the Sox complex involved in thiosulfate oxidation (*soxAXBYZ*) and of the rDsr system of elemental sulfur oxidation to sulfite (*dsrABCEFHEMPKJOL*) are present in the core genome. The genes encoding enzyme for direct (SoeABC) and indirect (Sat and AprAB) sulfite oxidation are also present in all species.

The core genome contains genes *hyaABC* encoding type I membrane-bound respiratory H_2_-uptake [NiFe]-hydrogenase, as well as *hypABCDEF* genes involved in hydrogenase maturation, which indicates the potential for lithotrophic growth in the presence of molecular hydrogen.

All members of the genus *Thiothrix* can assimilate carbon dioxide through the Calvin–Benson–Bassham cycle. The genes responsible for the regeneration and reduction of the substrate in the Calvin–Benson–Bassham cycle were found in the core genome. RuBisCO in the genus *Thiothrix* is represented by three types: IAq, IAc, and II. Previous pangenome analysis has shown that RuBisCO of the IAq type is common to all species, in contrast to IAc and II types [8]. However, the addition of ‘*Thiothrix winogradskyi*’ CT3^T^ into the pangenome analysis excluded type IAq RuBisCO from the core genome, since its genome does not contain this type of RuBisCO. All species, except for ‘*Ca*. Thiothrix anitrata’ A52, encode RuBisCO of type II, and all species except *T. fructosivorans* Q^T^, *T. lacustris* BL^T^, and MAG of *Thiothrix* sp. 207 encode type IAc (Figure 5).

As for phosphoribulokinase in the genus *Thiothrix*, the *prk* gene is represented by two phylogenetically different copies. The first copy has a high level of sequence similarity among all representatives of *Thiothrix* and was found in the core genome. The second copy of *prk* is also present in all *Thiothrix*; however, the amino acid sequence identity with the first type of phosphoribulokinase is below the 70% threshold used to determine the genes of the core genome. The genes for the tricarboxylic acid cycle, Embden–Meyerhof–Parnas pathway, the glyoxylate cycle, and phosphorus metabolism are mainly included in the core genome except for phosphate transport system permease (*pstB*), glucokinase (*glk*), dihydrolipoamide dehydrogenase of pyruvate dehydrogenase complex (*pdhD*), aconitate hydratase (*acnB*), and 6-phosphogluconolactonase (*pgl*), the level of similarity of which is below 70%. The genes of the first, second, and third complexes of the electron transport chain, as well as ATPase (complex V), are present in the core genome. The genes of the fourth respiratory chain complex were found in the core genome (*cox15*, *ccoO*, *ccoP*, *ccoG*, *ccoQ*, *ccoS*). Genes *cyoE*, *cox11*, *coxBACD*, *cydABCDX*, and *ccoN* are also present in all species but fell into the auxiliary genome due to higher sequence divergence. The cytochrome ubiquinol oxidase is present in all *Thiothrix* species except for ‘*Thiothrix winogradskyi*’ CT3^T^.

Pangenome analysis of the genus *Thiothrix* revealed that variable part of the genome that comprises genes responsible for anaerobic respiration in the presence of nitrogen compounds and nitrogen fixation. The distribution of genes responsible for different stages of nitrate reduction is irregular (Figure 5). The gene encoding nitrate reductase (Nar) is present in all *Thiothrix* species, except for ‘*Ca*. Thiothrix anitrata’ A52, ‘*Ca*. Thiothrix sulfatifontis’ KT, and *T. nivea* JP2^T^. Moreover, *T. lacustris* BL^T^, *T. litoralis* AS^T^, *T. fructosivorans* Q^T^, ‘*Thiothrix subterranea*’ Ku-5^T^, *T. caldifontis* G1^T^, and ‘*Thiothrix winogradskyi*’ CT3^T^ encode one form of NarG, *Ca.* Thiothrix moscovensis RT, and the MAG of *Thiothrix* sp. 207 encode another form, and *T. unzii* A1^T^ encodes both forms. Such a distribution pattern is probably the result of horizontal transfer with the subsequent loss of one of the two *narG* genes [8]. The *T. nivea* JP2^T^ genome contains genes for periplasmic nitrate reductase NapAB instead of Nar, while the MAG of *Thiothrix* sp. 207 contains both *nar* and *nap* genes. The gene for assimilatory nitrate reductase (*nasA*) was found in all members of the genus, except for ‘*Ca*. Thiothrix anitrata’ A52.

Nitrite could be reduced to ammonium by the assimilatory nitrite reductase NirBD, which is encoded in the genomes of all species, with the exception of ‘*Ca*. Thiothrix anitrata’ A52. The gene encoding dissimilatory nitrite reductase (*nirS*), involved in the reduction of nitrite to NO, was not found in ‘*Ca*. Thiothrix anitrata’ A52, ‘*Ca*. Thiothrix sulfatifontis’ KT, *T. nivea* JP2^T^, *Ca.* Thiothrix moscovensis RT, ‘*Thiothrix subterranea*’ Ku-5^T^, or *T. lacustris* BL^T^. Nitric oxide reductase NorBC catalyses the reduction of NO to N_2_O. ‘*Ca*. Thiothrix anitrata’ A52, ‘*Ca*. Thiothrix sulfatifontis’ KT, *T. nivea* JP2^T^, ‘*Thiothrix subterranea*’ Ku-5^T^, and *T. lacustris* BL^T^ lacked these genes. Analysis of the MAG of *Thiothrix* sp. 207 revealed the *nosZ* gene encoding nitrous oxide reductase, which enables the reduction of N_2_O to molecular nitrogen. This gene was most likely acquired via horizontal transfer from other gammaproteobacteria. *Thiothrix* sp. 207 therefore could be capable of complete denitrification.

Classical amination is represented by the genes for glutamine synthetase (*glnA*), glutamate synthase (*gltBD*), and aspartate aminotransferase (*aspB*) in the core genome.

The whole set of genes for the Nif-cluster (*nifASUBNXX2YB2ENQVWMHDKZTO*) was found in *T. litoralis* AS^T^, *T. caldifontis* G1^T^, *T. unzii* A1^T^, the MAG of *Thiothrix* sp. 207, *Ca.* Thiothrix moscovensis RT, ‘*Thiothrix subterranea*’ Ku-5^T^, and *T. nivea* JP2^T^ (Figure 5). *T. lacustris* BL^T^, ‘*Ca*. Thiothrix anitrata’ A52, ‘*Thiothrix winogradskyi*’ CT3^T^, and *T. fructosivorans* Q^T^ lack a set of genes for the *nif*-cluster, indicating their inability to assimilate N_2_. ‘*Ca*. Thiothrix sulfatifontis’ KT does not contain the *nifHDK* operon encoding two components of nitrogenase, which calls into question its capability for nitrogen fixation (Figure 5).

Previously, respiration in the presence of thiosulfate as a terminal electron acceptor was not described for any *Thiothrix* species. Genome analysis revealed that all analyzed members of the genus *Thiothrix*, except for *Ca.* Thiothrix moscovensis RT and ‘*Thiothrix winogradskyi*’ CT3, contain *phsABC* genes encoding thiosulfate reductase of the molybdopterin oxidoreductase family (Figure 5). Verification of genomic data, in particular the determination of the ability for anaerobic respiration in the presence of thiosulfate as an electron acceptor and lactate as an electron donor, as well as the determination of the products of the conversion of thiosulfate to hydrogen sulfide and sulfite, proved that most representatives of the genus *Thiothrix* are capable of dissimilatory thiosulfate reduction. A high sequence similarity of the deduced amino acid sequences of PhsA (>80%) most likely indicates the loss of the *phs* genes in *Ca.* Thiothrix moscovensis RT and ‘*Thiothrix winogradskyi*’ CT3 [17].

Thus, representatives of the genus *Thiothrix* differ significantly in their ability for anaerobic respiration in the presence of various electron acceptors (Figure 5). Such metabolic diversity can serve to adapt to changing environmental conditions.

The study of the composition of the species-specific genes in the genus *Thiothrix* has expanded our understanding of the metabolic potential of some species. Although most of such genes were predicted to encode hypothetical proteins with unknown functions, the presence of *nosZ* genes in the MAG of *Thiothrix* sp. 207 indicates that it could reduce N_2_O to molecular nitrogen, a trait absent in other *Thiothrix* species.

## 4. Environmental Distribution and Ecological Functions

The first species of the genus *Thiothrix* were isolated from sulfidic springs, and for a long time, they were considered limited to such habitats. A common feature of these habitats is the low organic content and the simultaneous presence of hydrogen sulfide and oxygen, which is consistent with the ability of all members of the genus *Thiothrix* to grow chemolithoautotrophically, obtaining energy from the aerobic oxidation of reduced sulfur compounds.

Later, 16S rRNA gene sequences assigned to *Thiothrix* were found by molecular profiling of microbial communities from organic-rich habitats, including the rhizosphere of the aquatic plant *Eichhornia crassipes* [21], the larvae of an aquatic insect *Chironomus ramosus* [22], and in activated sludge from wastewater treatment plants and laboratory bioreactors simulating wastewater treatment systems [23,24,25,26,27]. In all of the above works, the taxonomic composition of the microbiome of a plant, animal, or bioreactor was studied. In some of them [22,23], the community metagenome was also sequenced and the genes of certain metabolic pathways were analyzed. However, in these communities, the relative abundance of *Thiothrix* did not exceed a few percent. It can be assumed that in these habitats, the source of hydrogen sulfide can be either microbial sulfate reduction or decomposition of protein substrates under anaerobic conditions. It is possible that, in these ecosystems, *Thiothrix* grow organotrophically, receiving energy from the oxidation of organic substances rather than hydrogen sulfide. Although no cultured isolates have been obtained from these habitats to date, analysis of *Thiothrix* MAGs obtained from metagenomes of activated sludge from wastewater treatment plants did not reveal any significant peculiarities in their metabolic capabilities [17]. Representatives of the genus *Thiothrix* are much more widespread in nature than previously thought, but they become the dominant group only under specific conditions of organic-poor waters, simultaneously containing hydrogen sulfide and oxygen.

Bacteria of the genus *Thiothrix* form ectosymbiosis with marine and freshwater amphipods. For the first time, ectosymbiosis of bacteria from the genus *Thiothrix* with invertebrates was found in marine amphipods, *Urothoe poseidonis* [28]. However, the best studied is the ectosymbiosis of bacteria from the genus *Thiothrix* with freshwater amphipods of the genus *Niphargus*. Examples of such symbiosis were found in sulfide-rich waters in underground caves in Italy and Romania [29,30]. It has been suggested that a similar symbiosis may be much more widespread in European groundwater [30]. *Thiothrix* filaments are attached to the base of hairs on amphipod appendages. Bacteria from the genus *Thiothrix* have not been shown to be involved in the detoxification of hydrogen sulfide for the host organism, so the ecological role of this ectosymbiosis is not yet clear [31].

## 5. Comparison with Phylogenetically Related Genera *Thiolinea* and *Thiofilum*

The genera *Thiolinea* and *Thiofilum* are the closest phylogenetic relatives of the genus *Thiothrix* (Figure 3), initially described as members of that genus. Determination of genome sequences for representatives of these two genera (‘*Thiolinea eikelboomii*’, *Thiolinea disciformis*, and *Thiofilum flexile*) allowed a comparative genome-based analysis of the main metabolic pathways.

First of all, members of all three genera have the same enzyme systems for the oxidation of sulfur and hydrogen sulfide under aerobic conditions. Members of the genus *Thiolinea*, such as *Thiothrix*, are capable of autotrophic carbon fixation through the Calvin cycle. Their genomes contain genes encoding RuBisCO types IAq and II [15]. The phylogenetically more distant *Thiofilum flexile* lacks the RuBisCO genes and therefore is capable of only lithoheterotrophic but not lithoautotrophic growth.

Like different *Thiothrix* species, the genomes of *Thiolinea* and *Thiofilum* encode different sets of genes to reduce nitrogen compounds. In the genomes of *Thiofilum flexile* and *Thiolinea disciformis*, genes for denitrification systems (*narG*, *nirS*, *norBC*, *nosZ*) were not found, while the ‘*Thiolinea eikelboomii’* genome contains only genes for dissimilatory Nar-type nitrate reductase. Unlike most *Thiothrix* species, none of the members of *Thiolinea* and *Thiofilum* have genes of the *nif*-cluster, which enables nitrogen fixation.

As noted above, one of the features of the genus *Thiothrix* is the presence of a FAD-dependent membrane-bound malate:quinone oxidoreductase instead of the typical NAD-dependent cytoplasmic malate dehydrogenase. The *mqo* genes are also present in the genomes of *Thiolinea* and *Thiofilum flexile*, but the latter also contains the *mdh* gene. It is likely that the acquisition of *mqo* occurred in a common ancestor of all three genera, and *mdh* was subsequently lost in the *Thiolinea*/*Thiothrix* lineage.

## 6. Conclusions

A pangenome analysis of the genus *Thiothrix* revealed that the genes of dissimilatory sulfur metabolism are the most conserved part of the genome. The ability to carry out the interconversions of reduced sulfur compounds is probably a key characteristic for these bacteria, which once again indicates their lithotrophic type of metabolism.

*Thiothrix* thrives as a surface-attached organism in running water, which contains H_2_S and O_2_ in fairly high concentrations. In such habitats, *Thiothrix* filaments exist under conditions in which the concentrations of these substances (oxygen and hydrogen sulfide) could change significantly in a short period of time. As a rule, the O_2_ concentration in *Thiothrix* growth zones can vary from full to 10% of saturation. In such frequently changing environmental conditions, bacteria have developed an adaptation strategy through changes at the genomic level.

Significant differences in the inventory of genes for nitrogen metabolism, RuBisCO gene profiles [8], and *phsABC* thiosulfate reductase genes, and the discovery of new species-specific genes (*nosZ*), indicate a dynamic process of gene acquisition via horizontal transfer and lineage-specific gene losses that allows bacteria to adapt to variable environmental conditions. Thus, thiosulfate and oxidized forms of nitrogen, which can be used as electron acceptors during anaerobic respiration, are not always available in different ecosystems and can be interchanged; the presence of various RuBisCO forms can be explained by the adaptation to different concentrations of CO_2_ and O_2_, which vary in a wide range.

## Figures and Tables

**Figure 1 ijms-23-09531-f001:**
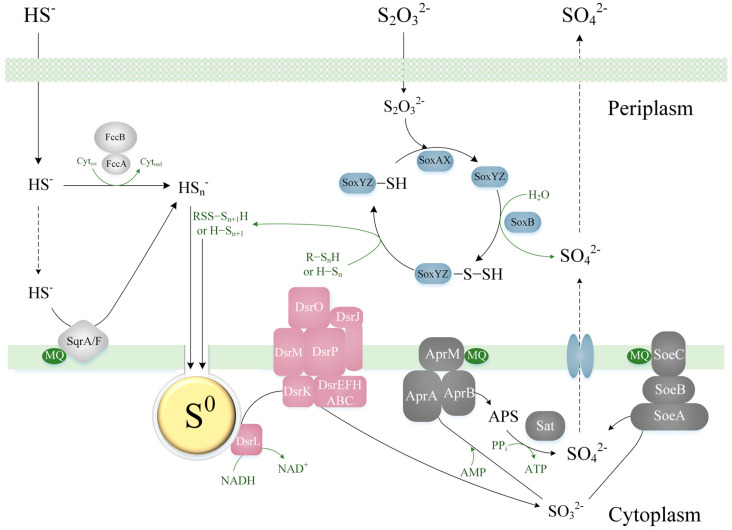
Scheme of dissimilation sulfur metabolism of the genus *Thiothrix*. FccAB, flavocytochrome *c*-sulfide dehydrogenase; SqrA/F, sulfide:quinone oxidoreductase; SoxAXBYZ, SOX multienzyme system; MQ, menaquinone; rDsrABCEFHEMPKJOL, dissimilatory sulfite reductase; AprABM, APS reductase; Sat, ATP sulfurylase; SoeABC, membrane-bound cytoplasmic sulfite:quinone oxidoreductase; S^0^, sulfur globule; APS, adenosine 5′-phosphosulfate; R−S_n+1_H, thiol compound; H−S_n+1_, polysulfide.

**Figure 2 ijms-23-09531-f002:**
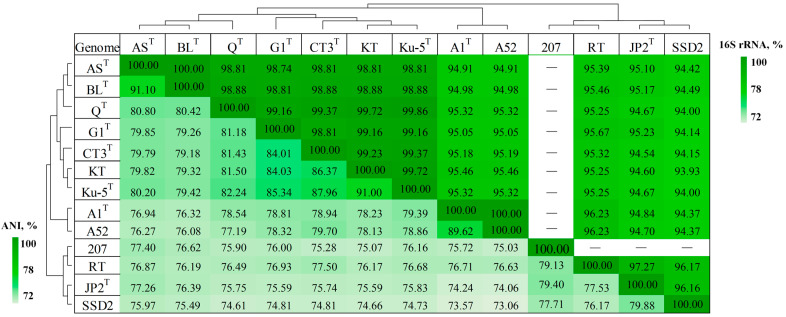
Heatmap of 16S rRNA gene sequence similarity and pairwise ANI values (%) for *Thiothrix* genomes. *T. lacustris* BL^T^, (GCF_000621325.1); *Thiothrix litoralis* AS^T^ (GCF_017901135.1); ‘*Thiothrix subterranea*’ Ku-5^T^ (GCF_016772315.1); ‘*Ca.* Thiothrix sulfatifontis*’* KT (GCA_022828425.1); *T. caldifontis* G1^T^ (GCF_900107695.1); ‘*Thiothrix winogradskyi*’ CT3^T^ (GCF_021650945.1); *T. fructosivorans* Q^T^ (GCA_017349355.1); *T. unzii* A1^T^ (GCA_017901175.1); ‘*Ca*. Thiothrix anitrata*’* A52 (GCF_017901155.1); *Ca.* Thiothrix moscovensis RT (GCA_016292235.1); *T. nivea* JP2^T^ (GCF_000260135.1); *Ca.* Thiothrix singaporensis SSD2 (GCA_013693955.1); MAG of *Thiothrix* sp. 207 (GCA_018813855.1). Note that 16S rRNA gene is missing in MAG of *Thiothrix* sp. 207.

**Figure 3 ijms-23-09531-f003:**
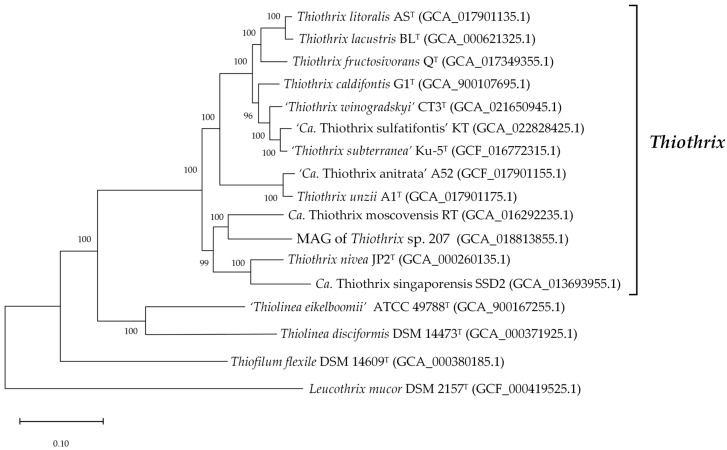
Genome-based phylogenetic tree of type strains of *Thiothrix* species. The GenBank assembly accession numbers are listed after the genome names. The internal branching support levels assessed by the Bayesian test in PhyML are specified in nodes. The genome of *Leucothrix mucor* DSM 2157^T^ was used for tree rooting.

**Figure 4 ijms-23-09531-f004:**
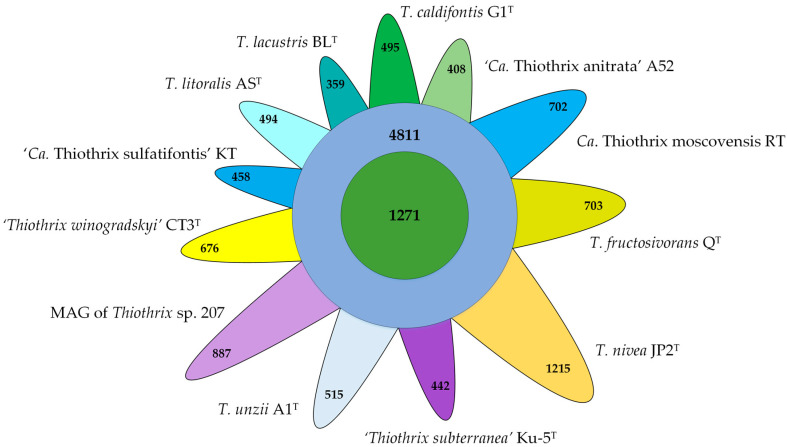
Venn diagram demonstrating the distribution of common (green), additional (purple), and unique (different colors) genes among analyzed *Thiothrix* species.

**Figure 5 ijms-23-09531-f005:**
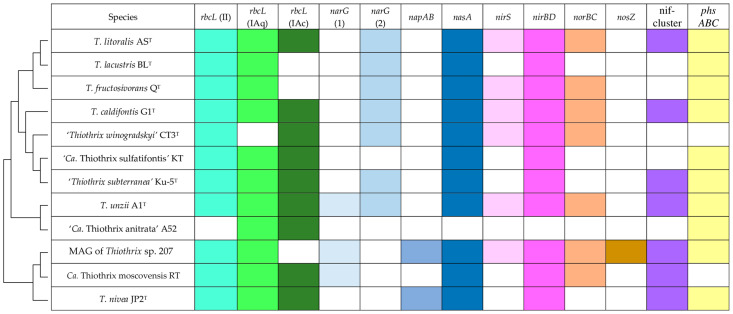
Distribution of genes related to nitrogen metabolism and carbon fixation in *Thiothrix* genomes. Genes: *rbcL*, the large subunit of RuBisCO; *narG*, membrane-bound nitrate reductase; *napAB*, periplasmic nitrate reductase; *nasA*, assimilatory nitrate reductase; *nirS*, dissimilatory nitrite reductase; *nirBD*, ammonia-forming assimilatory nitrite reductase; *norBC*, nitric oxide reductase; *nosZ*, nitrous oxide reductase; *nif*-cluster, nitrogenase genes; *phsABC*, thiosulfate reductase. Colored blocks indicate the presence of a gene.

**Table 1 ijms-23-09531-t001:** The general properties of *Thiothrix* genomes.

Species	Genome Assembly	Isolate or MAG	Size (MB)	Contigs	G + C Content (mol %)	Protein-Coding Genes	16S rRNA Genes	tRNA Genes	Plasmids *
*‘T. winogradskyi’* CT3^T^ (DSM 12730^T^)	GCA_021650935.1	Isolate	4.38	3	51.4	4292	5	66	2
‘*Ca.* Thiothrix sulfatifontis’ KT	GCA_022828425.1	MAG	3.69	1	51.5	3729	2	47	NA
*T. lacustris* BL^T^ (DSM 21227^T^)	GCF_000621325.1	Isolate	3.72	56	51.3	3537	2	40	U
*T. litoralis* AS^T^ (DSM 113264^T^)	GCF_017901135.1	Isolate	4.28	1	52.8	4045	3	44	0
*‘T. subterranea’* Ku-5^T^ (VKM B-3544^T^)	GCF_016772315.1	Isolate	4.08	4	51.1	3885	4	46	3
*T. caldifontis* G1^T^ (DSM 21228^T^)	GCF_900107695.1	Isolate	3.94	72	50.6	3752	1	42	U
*T. unzii* A1^T^ (ATCC 49747^T^)	GCA_017901175.1	Isolate	3.72	8	50.8	3626	2	45	7
*T. nivea* JP2^T^ (DSM 5205^T^)	GCF_000260135.1	Isolate	4.69	15	54.9	4327	2	44	U
*T. fructosivorans* Q^T^ (ATCC 49748^T^)	GCA_017349355.1	Isolate	4.36	6	51.3	3616	2	44	5
*Ca.* Thiothrix moscovensis RT	GCA_016292235.1	MAG	3.69	78	53.6	3483	1	38	NA
*Ca.* Thiothrix singaporensis SSD2	GCA_013693955.1	MAG	4.54	1	55.6	4097	2	43	NA
‘*Ca.* Thiothrix anitrata’ A52	GCF_017901155.1	MAG	3.55	1	50.1	3387	2	45	NA
*Thiothrix* sp. 207	GCA_018813855.1	MAG	3.93	136	54.6	3675	0	40	NA

* U, unknown because the assembly consisted of multiple contigs; NA, not applicable for MAGs.

## Data Availability

Not applicable.

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
