# Peer review of "History of the Study of the Genus Thiothrix: From the First Enrichment Cultures to Pangenomic Analysis"

_ijms, 2022, doi:10.3390/ijms23179531_

Round 1
Reviewer 1 Report
The paper provides new knowledge concerning the thiotrix genus and a new candidate species. The main advantages of the paper are that the authors are very knowledgeable about the subject and provide with precise and specialised insights concerning this bacterial genus. The quality of the paper is quite high, supported by the detailed literature review, and it can further improved by the visualisation and the additional small analyses proposed. Overall the authors did a very good job, and the paper can be improved making it a quite interesting paper that will receive citations over time. However, as mentioned before some more visualisation and analysis could greatly improve the quality and further highlight the main points of the paper. Moreover, the materials and method section is not provided, which makes the review process incomplete. I apologise if i missed something concerning the materials and methods of the specific journal and the policy it has concerning this matter. My suggestion is that the paper can be accepted after minor revisions
English corrections
line 39 : The hydrogen sulfide concentration
line 41 : forms
line 41 : litre
Figure 1 : periplasm and cytoplasm
line 66 : are differentiated
line 66 : based on
line 68 : final invalidity
line 70 : for a representative
line 75 : of the XX century
line 80 : in the genus
line 86 : delete "them"
line 103: genera,
line 108: Boden and Scott based on...
line 246: Nitric oxide reductase CnorBC catalyses the reduction of The reduction of NO to N2O
line 282: ,and ... time, (delete as)
line 294: were studied. In some of them, ...
line 303: delete apparently
line 330: to reduce nitrogen
Comments on the figures, methodology and scientific soundness
Concerning the methodology of the paper (the materials and methods were not yet available so this must be checked also at the second round?)
1. Figure 5 showing specific differences should be improved, since this is maybe the main highlight of the paper. An anvio figure or a circos plot showing the location or presence absence of specific genes along with the genome completeness for each species so that it is very clear
2. concerning the specific genes found, are there any operons connected to them ?
Author Response
Dear Reviewer,
Thank you very much for your questions and recommendations. All revisions was highlighted using the "Track Changes" function in Microsoft Word.
English corrections
line 39 : The hydrogen sulfide concentration
RE: We corrected «The concentration of hydrogen sulfide» to «The hydrogen sulfide concentration» in line 37.
line 41 : forms
RE: We corrected «form» to «forms» in line 39.
line 41 : litre
RE: We corrected «liter» to «litre» in line 39.
Figure 1 : periplasm and cytoplasm
RE: We have made changes in Figure 1 in accordance with your recommendations.
line 66 : are differentiated
RE: We have adjusted the sentence in accordance with your recommendations in line 65.
line 66 : based on
RE: We have adjusted the sentence in accordance with your recommendations in line 65.
line 68 : final invalidity
RE: We corrected «finally invalidity» to «final invalidity» in line 67.
line 70 : for a representative
RE: We corrected «for representative» to «for a representative» in line 69.
line 75 : of the XX century
RE: We have made changes as per your recommendation in line 74.
line 80 : in the genus
RE: We corrected «into the genus» to «in the genus» in line 79.
line 86 : delete "them"
RE: We deleted «them» in line 85.
line 103: genera,
RE: We have made the change as per your recommendation in line 102.
line 108: Boden and Scott based on...
RE: We have adjusted the sentence in accordance with your recommendations in line 107.
line 246: Nitric oxide reductase CnorBC catalyses the reduction of The reduction of NO to N2O
RE: We have adjusted the sentence in accordance with your recommendations in line 246.
line 282: ,and ... time, (delete as)
RE: We deleted «as» in line 283.
line 294: were studied. In some of them, ...
RE: We have adjusted the sentence in accordance with your recommendations in line 294.
line 303: delete apparently
RE: We deleted «apparently» in line 303.
line 330: to reduce nitrogen
RE: We corrected « for the reduction of nitrogen compounds» to «to reduce nitrogen» in line 330.
Comments on the figures, methodology and scientific soundness
Concerning the methodology of the paper (the materials and methods were not yet available so this must be checked also at the second round?)
RE: Thank you for your remark! We have checked necessary sections for MDPI Article Types
(https://www.mdpi.com/about/article_types). The structure of Review Article should not include
Materials and Methods. It can include an Abstract, Keywords, Introduction, Relevant Sections,
Discussion, Conclusions, and Future Directions.
- Figure 5 showing specific differences should be improved, since this is maybe the main highlight of the paper. An anvio figure or a circos plot showing the location or presence absence of specific genes along with the genome completeness for each species so that it is very clear
RE: The purpose of Figure 5 was to illustrate the distribution of genes involved in nitrogen metabolism and autotrophic carbon fixation across the genomes. We suppose that the present form of presentation (a table combined with phylogenetic scheme) is the most appropriate. Anvi’o package was designed for analysis and visualization of metagenomic assemblies and this is not our case. Most of analysed genomes represented isolates (and are apparently 100% complete) and only 4 are high quality MAGs with more than 98% completeness. We added information about completeness of the MAGs to the “Pangenome analysis” section.
- concerning the specific genes found, are there any operons connected to them ?
RE: At the moment, there is little information regarding the operon organization of the genome for the Thiothrix group. This is our future issue. Nevertheless, the available information about Calvin-Benson-Bassham cycle genes organization in other organisms and the available genomic data for representatives of Thiothrix suggest that the genes for the key enzymes RuBisCO (rbcL-IAc, IAq, II) and phosphoribulokinase in Thiothrix are similarly located in the cbb operon.
A similar picture is seen for denitrification and dissimilation nitrate reduction. The system genes responsible for the reduction of nitrates (narGHI or napAB), nitrites (nirBD), NO to N2O norBC are organized in the corresponding operons. nif-cluster is a system operating under the strict control of the operon system (nifBHDKENX), which is also seen in Thiothrix members.
Sincerely yours,
Margarita Grabovich

Reviewer 2 Report
The proposed review paper concerns the history of the study of Thiothrix genus. The paper is well written, although some small modifications are required. It could be accepted with minor revision.
The proposed modifications are the following:
Abstract: if it is presented as a "History" paper, the abstract is too much dedicated to the Pangenomic Results (almost 2/3 of abstract). I know that they are the more recent and interesting, but in these paragraph I suggest to not describe all those detailed results, try to be more general just highlighting the more important.
Line 53-54: the acronym SOX is still not explicated. In lines 55-56 pay attention to the brackets.
Paragraph 3, line 164: from this point, why did you exclude Ca. T. singaporensis SSD2 from Venn diagram and the following evaluation? It is not clear.
Line 182: SQR is well explained at line 52 (sulfide:quinone REDUCTASE), here you should delete "oxidoreductase".
Line 222: why "both"? The complex described is one (the fourth), while the genes are six.
Line 335: you could delete "characteristic"
Line 387: you missed a "o" in the word "conservation"
Author Response
Dear Reviewer,
Thank you very much for your questions and recommendations. All revisions was highlighted using the "Track Changes" function in Microsoft Word.
The proposed review paper concerns the history of the study of Thiothrix genus. The paper is well written, although some small modifications are required. It could be accepted with minor revision.
The proposed modifications are the following:
Abstract: if it is presented as a "History" paper, the abstract is too much dedicated to the Pangenomic Results (almost 2/3 of abstract). I know that they are the more recent and interesting, but in these paragraph I suggest to not describe all those detailed results, try to be more general just highlighting the more important.
RE: We have adjusted the abstract according to your recommendations.
Line 53-54: the acronym SOX is still not explicated. In lines 55-56 pay attention to the brackets.
RE: We added the explication for SOX in line 52.
Paragraph 3, line 164: from this point, why did you exclude Ca. T. singaporensis SSD2 from Venn diagram and the following evaluation? It is not clear.
RE: MAG of “Ca. Thiothrix singaporensis” SSD2 was obtained upon metagenome sequencing using only Oxford Nanopore technique. Its sequence contained multiple frameshifts complicating correct identification of genes and therefore this MAG was excluded from the analysis. We added an explanation.
Line 182: SQR is well explained at line 52 (sulfide:quinone REDUCTASE), here you should delete "oxidoreductase".
RE: We corrected «reductase» to «oxidoreductase» in line 50. We deleted «oxidoreductase» in line 184.
Line 222: why "both"? The complex described is one (the fourth), while the genes are six.
RE: We deleted «both» in line 225. We meant that all of the genes encoding the fourth complex, only six genes are found in the core genome of representatives of the genus Thiothrix.
Line 335: you could delete "characteristic"
RE: We deleted «characteristic» as per your recommendation in line 335.
Line 387: you missed a "o" in the word "conservation"
RE: We corrected the word "conservation" in line 387.
Sincerely yours,
Margarita Grabovich

This manuscript is a resubmission of an earlier submission. The following is a list of the peer review reports and author responses from that submission.